# Genome-wide gene-based analyses of weight loss interventions identify a potential role for *NKX6.3* in metabolism

Armand Valsesia[1], Qiao-Ping Wang[2,3], Nele Gheldof[1], Jérôme Carayol [1], Hélène Ruffieux[1,4], Teleri Clark[2], Victoria Shenton[2], Lisa J. Oyston[2], Gregory Lefebvre[1], Sylviane Metairon [1], Christian Chabert[1], Ondine Walter[1], Polina Mironova[1], Paulina Lau[5], Patrick Descombes [1], Nathalie Viguerie [6], Dominique Langin [6,7], Mary-Ellen Harper [8], Arne Astrup [9], Wim H. Saris [10], Robert Dent[4], Greg G. Neely[2] & Jörg Hager[1]

Hundreds of genetic variants have been associated with Body Mass Index (BMI) through genome-wide association studies (GWAS) using observational cohorts. However, the genetic contribution to efficient weight loss in response to dietary intervention remains unknown. We perform a GWAS in two large low-caloric diet intervention cohorts of obese participants. Two loci close to *NKX6.3/MIR486* and *RBSG4* are identified in the Canadian discovery cohort ($n = 1166$) and replicated in the DiOGenes cohort ($n = 789$). Modulation of *HGTX* (*NKX6.3* ortholog) levels in *Drosophila melanogaster* leads to significantly altered triglyceride levels. Additional tissue-specific experiments demonstrate an action through the oenocytes, fly hepatocyte-like cells that regulate lipid metabolism. Our results identify genetic variants associated with the efficacy of weight loss in obese subjects and identify a role for *NKX6.3* in lipid metabolism, and thereby possibly weight control.

[1] Nestlé Institute of Health Sciences, 1015 Lausanne, Switzerland. [2] Functional Genomics group, Charles Perkins Centre, University of Sydney, 2006 Sydney, NSW, Australia. [3] School of Pharmaceutical Sciences (Shenzhen), Sun Yat-sen University, 510275 Guangzhou, China. [4] Ecole Polytechnique Fédérale de Lausanne (EPFL), 1015 Lausanne, Switzerland. [5] Ottawa Hospital Weight Management Clinic, The Ottawa Hospital, K1Y 4E9 Ottawa, ON, Canada. [6] Institute of Metabolic and Cardiovascular Diseases, INSERM, Paul Sabatier University, UMR 1048, Obesity Research Laboratory, University of Toulouse, 31432 Toulouse, France. [7] Department of Clinical Biochemistry, Toulouse University Hospitals, 31432 Toulouse, France. [8] Department of Biochemistry, Microbiology and Immunology, Faculty of Medicine, University of Ottawa, Ottawa K1H 8M5 ON, Canada. [9] Department of Nutrition, Exercise and Sports, Faculty of Science, University of Copenhagen, 2200 Copenhagen, Denmark. [10] Department of Human Biology, NUTRIM, School of Nutrition and Translational Research in Metabolism, Maastricht University Medical Centre+(MUMC+), 6211 Maastricht, The Netherlands. These authors contributed equally: Armand Valsesia, Qiao-Ping Wang. These authors jointly supervised this work: Greg Neely, Jörg Hager. Correspondence and requests for materials should be addressed to A.V. (email: Armand.Valsesia@rd.nestle.com)

Obesity is a world-wide issue and a major risk factor for cardiovascular disease, dyslipidemia, hypertension, insulin resistance and type 2 diabetes as well as cancer[1–3]. A recent report from the NCD-RisC network has shown the increasing prevalence of obesity and estimated that with current post-2000 trends, the global obesity frequency would surpass 18% in men and 21% in women by 2025[4].

Multiple studies have shown that weight loss through energy restricted dietary interventions improves metabolic dysfunction[5,6]. Nevertheless, a large inter-individual variability is observed regarding the capacity to lose weight and to maintain the lost weight[7,8]. Genome-wide association studies (GWAS) from the GIANT consortium have identified about 100 loci associated with body mass index (BMI) variability in the general population[9]. Those candidate obesity loci were investigated in two lifestyle interventions: the Diabetes Prevention Program (DPP)[10,11] and Look AHEAD[12,13]. In these candidate analyses, only one marker (MTIF3-rs1885988) was associated with degree of weight loss and none to weight regain tendancy[14]. To date, no genome-wide approach for weight loss success has been undertaken[15].

Here we present results from a genome-wide association (GWA) study for weight loss using two large low-caloric diet interventions: the Canadian Optifast900® meal replacement program[16] and the DiOGenes clinical trial[17,18]. Our analyses implement a gene-based GWAS to maximize statistical power[19,20]. One cohort is used for discovery and the second, for replication. Next, we perform Bayesian risk variant inference based on joint modeling of the GWA signals and large-scale epigenome annotation data to restrict the association signals to the most likely associated SNPs. Finally, we perform functional RNAi knockdown in *Drosophila melanogaster* to study the potential in vivo metabolic impact of the regional candidate genes.

Our study provides evidence for a weight loss locus on chromosome 8p11 and knock out experiments in *Drosophila melanogaster* suggest the *NKX6.3* gene in the region as a potential functional candidate.

## Results

**Cohort descriptions**. The Optifast900 cohort included both obese and severely obese subjects (mean BMI = 43.2 kg/m2 ± 0.3 standard error of the mean) and the DiOGenes cohort included overweight and obese participants (mean BMI = 34.5 kg/m2 ± 0.2). Clinical characteristics of the participants are available in Table 1. Upon a 5-week low calorie diet (LCD), participants lost on average 9.3% (11.3 kg) and 7.5% (7.5 kg) of initial body weight, respectively for the Optifast900 and DiOGenes participants. At baseline, Optifast900 participants were considered more insulin-

resistant than DiOGenes subjects (HOMA-IR = 4.16 ± 0.14 vs. 3.15 ± 0.10), as expected given the more severe obesity.

**Gene-based association studies**. We searched for SNPs associated with degree of weight loss using the largest cohort (Optifast900, n = 1166) as a discovery dataset. The smaller cohort (DiOGenes, n = 789) was then used for replication. To unravel associations using single-SNP approaches at genome-wide scale, very large cohorts are needed, but such sample size cannot be obtained in randomized clinical trials. To better extract association signals, we used a gene-based approach that enables to integrate individual SNP association signals into a locus-wise signal (see Methods). Upon gene-based analyses of the Optifast900 cohort, we identified 12 genes, corresponding to 6 distinct loci with nearby SNPs significantly associated with weight loss (genome-wide FDR < 0.05 see Table 2). A Manhattan plot of the gene-based GWA results is available in Fig. 1.

Next, replication of those loci was attempted using the DiOGenes cohort. Two out of the six loci were successfully replicated in the DiOGenes cohort (two-stage FDR < 0.05, Table 2): the *RBSG4* locus on chromosome 1q24 and the *MIR486/NKX6.3* locus on chromosome 8p11. Meta-analysis using random-effect modeling of the two cohorts also showed significant association for these loci (both at genome-wide levels and with a two-stage approach) with effect sizes that were consistent between the two cohorts. Regional plots for those two loci are shown in Fig. 2. The *MIR486* gene has two isoforms with similar coordinates, thus essentially the same SNPs were included in the gene-based analyses leading to very similar p-values for MIR486-1 and MIR486-2 (as seen in Table 2).

**Bayesian framework for risk variant inference**. We took advantage of the development of a recent Bayesian framework, RiVIERA-beta[21] to infer posterior probabilities of disease association (PPA) that were then used to rank the associated SNPs. Upon Bayesian modeling of SNPs with nominal p-values less than 1e-3 for the two replicated loci (*RBSG4* and *MIR486/NKX6.3*), we were able to restrict the list of candidate SNPs to four markers, with a possible regulatory impact. Figure 3 summarizes the results of the Bayesian modeling and presents the overlap between variants and epigenomic annotations. Table 3 provides the effect size (from single-SNP GWAs) for each identified marker. For the *RBSG4* locus (Fig. 3a), we identified three markers (rs873822, rs870879, rs1027493) significantly enriched in epigenome annotations. Those markers were in strong LD with each other (r2 > 90%) and thus any of those would tag the others. Those three markers are common variants (with Minor Allele Frequency (MAF) = 27%). For the *MIR486/NKX6.3* locus, the rs6981587 SNP (MAF = 34%) emerged as the most likely risk

**Table 1 Descriptive statistics for the two studies used in the analysis**

|  | OPTIFAST900 (n = 1166) | DIOGENES (n = 789) | *p*-value |
|---|---|---|---|
| Number of males (%) | 237 (26.58%) | 310 (33.95%) | – |
| Age at baseline (years) | 46.50 ± 0.32 | 41.37 ± 0.23 | p < 0.001 |
| BMI at baseline (kg/m²) | 43.17 ± 0.23 | 34.53 ± 0.19 | p < 0.001 |
| Weight at baseline (kg) | 121.66 ± 0.76 | 99.74 ± 0.67 | p < 0.001 |
| Weight after 5-week LCD (kg) | 110.31 ± 0.68 | 92.20 ± 0.61 | p < 0.001 |
| Change in weight during LCD (kg) | −11.35 ± 0.11 | −7.55 ± 0.11 | p < 0.001 |
| Change in weight during LCD (%) | −9.28 ± 0.06 | −7.51 ± 0.09 | p < 0.001 |
| Fasting glucose levels (mmol/L) at baseline | 5.71 ± 0.05 | 5.12 ± 0.03 | p < 0.001 |
| HOMA-IR at baseline | 4.16 ± 0.14 | 3.15 ± 0.10 | p < 0.001 |

The *p*-value was obtained from a two-sided *t*-test and assesses differences between the two cohorts
LCD: low caloric diet, HOMA-IR: homeostasis model assessment of insulin resistance

**Table 2 Top hits from the gene-based GWA**

| Gene | Chr | Start | Stop | OPTIFAST900 (n = 1166) | DIOGENES (n = 789) | Meta-analysis |
|------|-----|-------|------|------------------------|--------------------|---------------|
| *MIR486-2* | 8 | 41497961 | 41538025 | **1.7e-05 (0.043)** | **0.000618 (0.004)** | **1e-06 (1.2e-05)** |
| *MIR486* | 8 | 41497958 | 41538026 | **2.2e-05 (0.043)** | **0.000661 (0.004)** | **2e-06 (1.2e-05)** |
| *NKX6-3* | 8 | 41483828 | 41524878 | **2.4e-05 (0.043)** | **0.00185 (0.0074)** | **2.6e-05 (0.0001)** |
| *RBSG4* | 1 | 167124598 | 167185042 | **2e-05 (0.043)** | **0.011 (0.033)** | **3.8e-05 (0.00011)** |
| *DBNDD1* | 16 | 90051272 | 90106539 | **7e-06 (0.035)** | 0.0305 (0.073) | **0.0116 (0.02)** |
| *PTPRT* | 20 | 40681391 | 41838557 | **6e-06 (0.035)** | 0.1728 (0.28) | **0.0062 (0.012)** |
| *GAS8* | 16 | 90066036 | 90131379 | **7e-06 (0.035)** | 0.1958 (0.28) | 0.06909 (0.1) |
| *CCBE1* | 18 | 57078170 | 57384644 | **8e-06 (0.035)** | 0.2008 (0.28) | 0.000346 (0.00083) |
| *C16orf3* | 16 | 90075315 | 90116309 | **2.1e-05 (0.043)** | 0.2118 (0.28) | 0.1049 (0.14) |
| *URAHP* | 16 | 90086168 | 90134191 | **1.1e-05 (0.04)** | 0.2747 (0.33) | 0.1339 (0.16) |
| *OR5K2* | 3 | 98196524 | 98237475 | **7e-06 (0.035)** | 0.5784 (0.63) | 0.1578 (0.17) |
| *OR5K1* | 3 | 98168420 | 98209372 | **1.3e-05 (0.04)** | 0.8541 (0.85) | 0.3117 (0.31) |

This table present gene-based association *p*-values for the two cohorts and their meta-analysis. False-discovery rate (FDR) adjusted *p*-value is indicated within parenthesis. For the discovery cohort (Optifast900), the FDR is a genome-wide FDR. For the replication cohort (DIOGENES) and the meta-analysis results, the FDR is adjusted for a two-stage analysis. FDRs less than 5% are shown in bold.
Source data are provided as a Source Data file

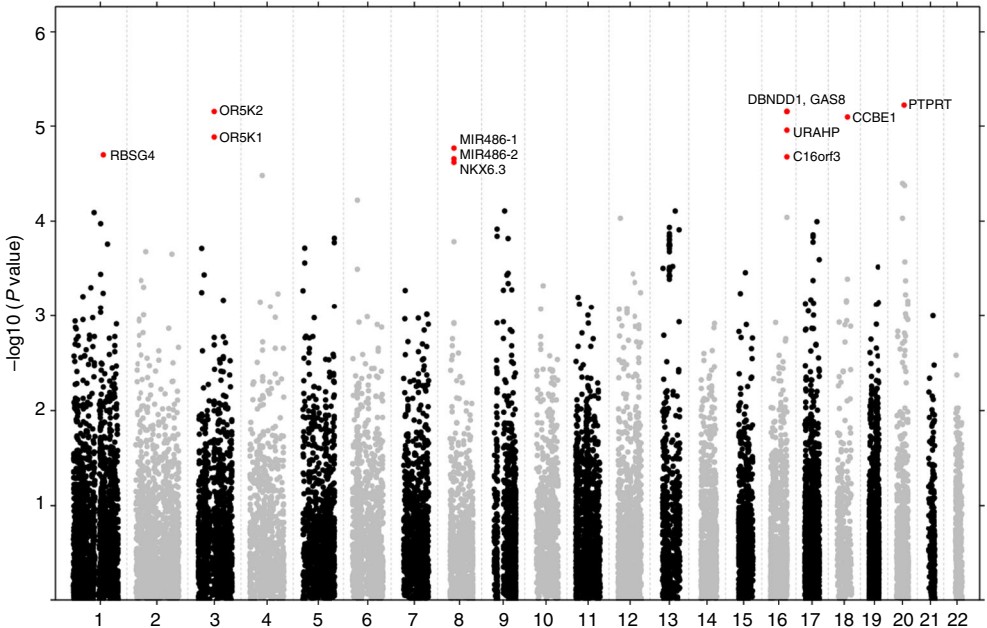

**Fig. 1** Manhattan plot: Gene-based association results for the discovery cohort (Optifast900), Highlighted genes (in red) correspond to loci with genome-wide significant association signals (FDR < 5%). Source data are provided as a Source Data file

variant. In this locus, five other SNPs had slightly lower *p*-values, yet none showed such enrichment in epigenome annotation marks (Fig. 3b). For the four SNPs in the two genomic loci, consistent effect sizes were observed between the two cohorts, as well as similar allele frequencies (Table 3). These analyses are useful to identify the most likely regulatory variants. However, they do not enable to infer which gene(s) may be impacted. Indeed, within the *MIR486/NKX6.3* locus (Fig. 3b), there are two other genes (*ANK1* and *AGPAT6*) that are in the vicinity of the top regulatory variant and that would also deserve functional follow-up.

**Functional assessment in *Drosophila melanogaster*.** To investigate a potential in vivo metabolic function for the genes around the risk variants, we used the fruit fly *Drosophila melanogaster*. Because *RBSG4* and *MIR486* are not conserved in the fly, we focused our analysis on *Ank/ANK1*, *HGTX/NKX6-3* and *CG3209/AGPAT6* and each gene was targeted using whole body RNAi knockdown (*Actin-Gal4*). There were no major

developmental effects for *Actin-Gal4 > UAS-Ank* and *Actin-Gal4 > UAS-CG3209* RNAi flies and we did not observe significant changes in TAG levels compared to their wild-type controls (Supplementary Fig. 2a). We also performed over-expression (OE) of *ANK1* using a whole-body driver (*Actin-Gal4*). There was no impact on triglyceride levels in the *Actin-Gal4 > UAS-ANK1 OE* animals compared to controls (Supplementary Fig. 2b). The majority of *Actin-Gal4 > UAS-HGTX* RNAi flies were developmentally lethal (>95% pupal lethality), however some animals did survive. From the viable *HGTX* knockdown flies (*Actin-Gal4 > UAS-HGTX* RNAi), we observed a significant reduction in triglyceride (TAG) level compared to controls (Fig. 4a). This finding was confirmed using a second RNAi hairpin (Supplementary Fig. 3a). Complete lethality of F1 progenies was observed in *HGTX* RNAi or *HGTX* overexpression flies using additional ubiquitous drivers *Tub-Gal4* and *DA-Gal4*. To bypass developmental effects of *HGTX* knockdown, we next performed an adult-specific inducible knockdown using the *TubGal80^{ts}* system. *Actin-Gal4; Gal80^{ts} >*

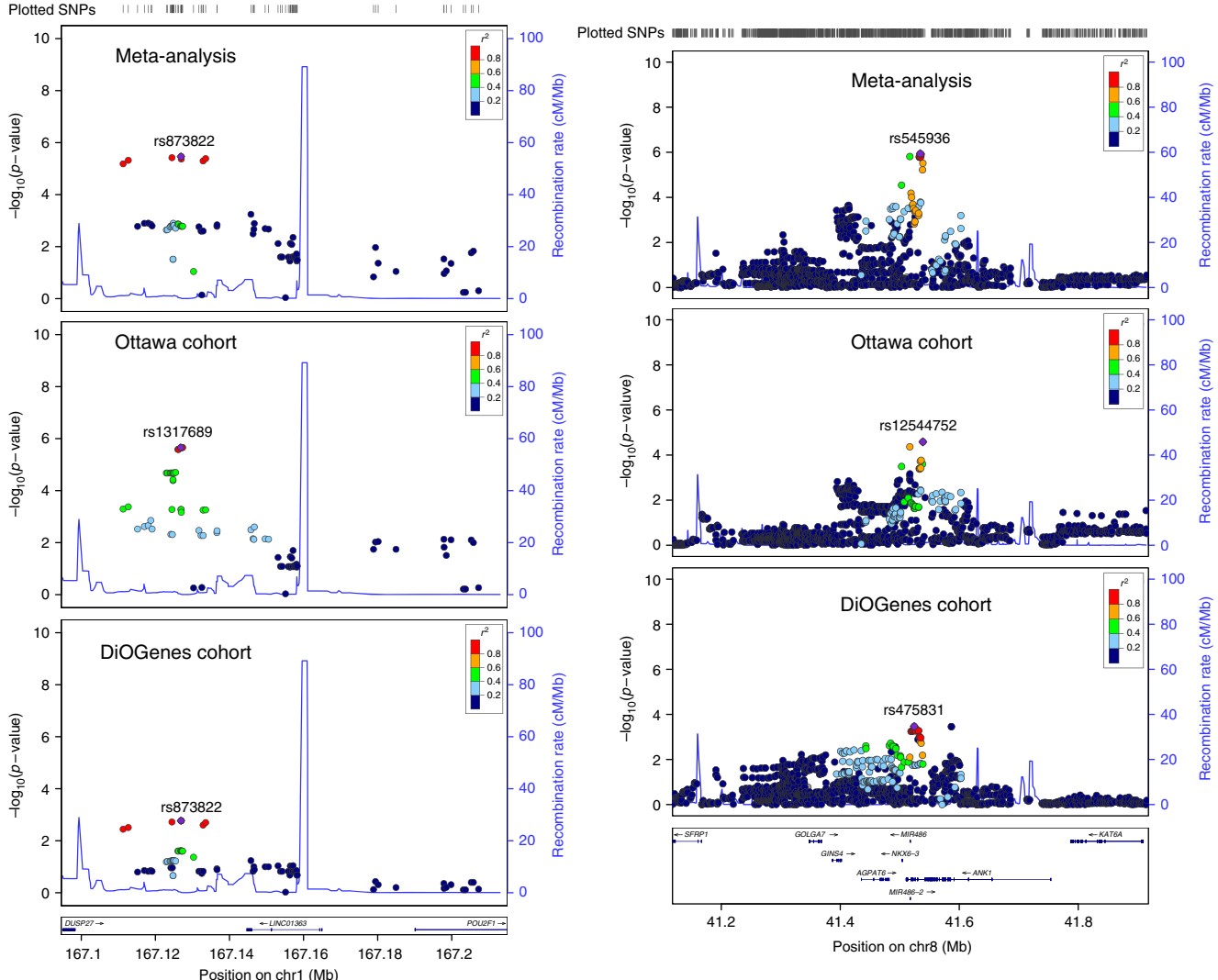

**Fig. 2** LocusZoom plots for the RBSG4 (LINC01363) and the MIR486/NKX6.3 loci. Left (right) panel corresponds to RBSG4 (MIR486/NKX6.3/ANK1). Panels from top to bottom correspond to results from the meta-analysis, the Optifast900 cohort and the DiOGenes cohort. Source data are provided as a Source Data file

*UAS-HGTX* RNAi animals were raised at 18 °C during developmental stage, which suppresses RNAi and then hatched flies were shifted to 29 °C for 6 days at which time RNAi is activated. Induced knock-down animals displayed a similar level of TAG reduction as constitutive *HGTX* knockdown animals compared to the parental controls (Fig. 4b). To confirm inducible RNAi knockdown efficiency, we used qPCR and observed approximate 60% reduction in *HGTX* mRNA levels (Fig. 4c). Further metabolic characterization of these inducible knockdown animals showed no significant difference in levels of glycogen (Supplementary Fig. 3b) or trehalose (Supplementary Fig. 3c), and body weight (Supplementary Fig. 3d), food intake (Supplementary Fig. 3e), and starvation response (Supplementary Fig. 3f) were also similar to the controls. Of note, *HGTX* inducible knockdown did not affect fly insulin-like peptide *Ilp2* or *Ilp5* expression but resulted in a decrease in *Ilp3* expression. However, we did not observe any difference in dilp3 expression at the protein level (Supplementary Fig. 4).

To further confirm the role of *HGTX* in regulation of TAG, we used inducible over-expression of *HGTX* in adults with mRNA expression ~9 times higher in *Actin-Gal4; Gal80ts > UAS-HGTX* OE animals (Fig. 4d) compared to the parental controls. *HGTX*

over-expression led to a mild reduction in TAG (Fig. 4e). No significant impact was observed for *Ilp2, 3* and 5 mRNA expression or dilp3 protein levels (Supplementary Fig. 5).

To find the specific tissue in which *HGTX* acts, we carried out tissue specific *HGTX* RNAi targeting expression in the fat body (*Ppl-Gal4*), muscle (*Mef2-Gal4*), brain (*nSyb-Gal4*) or oenocytes (*Oeno-Gal4*). We found that only oenocyte-specific knock down of *HGTX* resulted in a significant reduction in TAG compared to parental controls (Fig. 4f). Conversely, oenocyte-specific over-expression of *HGTX* resulted in a significant increase in TAG (Fig. 4g). Together, our data supports a role for HGTX/NKX6.3 acting in the fly oenocyte to regulate triglyceride content in vivo.

## Discussion

Here we describe a genome-wide association for weight loss. We used two independent weight loss cohorts: a Canadian Optifast900 cohort (n = 1166) for discovery and the pan-European DiOGenes cohort (n = 789) for replication.

Recent analysis[14] of the Look AHEAD and DPP cohorts only focused on 91 established obesity loci[9,22] and found association of only one of the loci with weight loss and regain after three years,

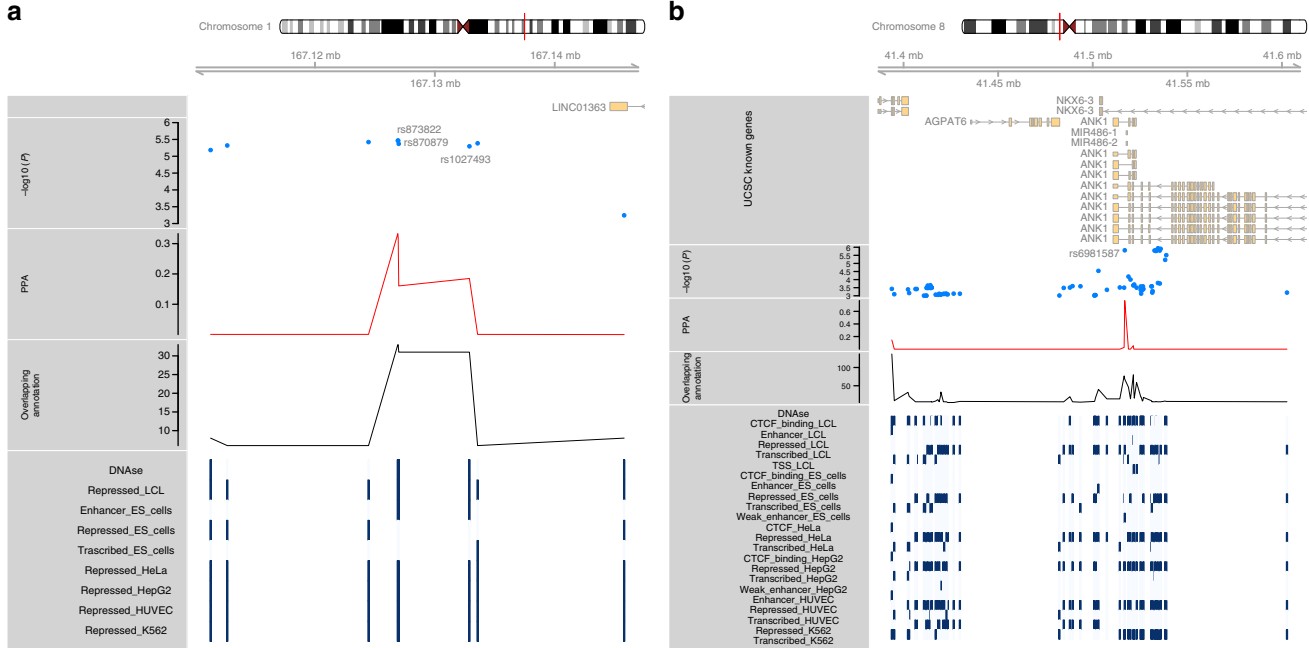

**Fig. 3** Bayesian risk variant inference for SNPs within the RBSG4 and the MIR486/NKX6.3 loci. Panel (**a**) corresponds to RBSG4 (LINC01363) and panel (**b**) to MIR486/NKX6.3. Tracks from top to bottom display: (1) gene track, (2) −log10 association p-values from the meta-analysis, (3) posterior probability of disease association (PPA), (4) the number of overlapping epigenomic annotation at a given marker, (5) the detail of the epigenomic marks that overlap a given variant. The risk inference analyses are limited to variants with p-values smaller than $10^{-3}$; all other variants are not displayed. Due to a large number of DNase annotation tracks (>100 for specific variants), these tracks have been reduced to a single one (referred as "DNase"). Source data are provided as a Source Data file

namely the *MTIF3*-rs1885988 SNP. Analysis of this SNP in our two cohorts did not confirm this association (meta-analysis p-value = 0.52). This could be due to differences between the behavior-based and low-caloric dietary interventions as well as the length/type of the intervention in the different studies. Analyses of the remaining 90 BMI loci in our data confirmed the lack of association of these loci with weight loss (all loci had FDR > 10%); further highlighting the need to investigate the genetic contribution to weight loss in well-controlled weight loss studies (i.e. with similar intervention design).

GWAs of complex, polygenic traits (such as metabolic diseases) require very large sample sizes to unravel common variants with low/moderate effects. Such large sample sizes are typically achieved with meta-analyses of numerous observational studies. For illustration, the latest GWAs from the GIANT consortium[23] used a sample size greater than 700,000 and identified 716 variants representing about 5% of BMI variation. Such sample sizes are not achievable for clinical intervention studies. However, over the past few years, the development of new statistical methods has enabled to increase statistical power using multi-marker analyses. We thus used a well-established gene-based strategy[20] and took advantage of recent improvements for single-SNP analyses by using a Bayesian linear-mixed effect model[24]. Gene-based approaches enable to combine association p-values from individual SNPs into a single locus (e.g. gene-level) p-value. This leads to several benefits. First, statistical evidence is strengthened by integrating association p-values from markers located within a same region. Some of those markers may already display association signals close to genome-wide significance thresholds, yielding a more extreme combined p-value. These approaches account for LD relationships between markers by using resampling approaches (e.g. Monte Carlo simulations). The second benefit from gene-based approaches pertains to a significant reduction of the multiple testing burden (as the number of tests is

about 20,000 regions instead of millions of SNPs). Finally, it has been previously discussed that gene-based approaches are less prone to spurious associations caused by population stratification compared to single-SNP or haplotype-based analyses[19,25].

By using such a combination of tools, we identified six different loci with genome-wide FDR < 5% in the discovery phase. Two of those loci (*RBSG4* and *MIR486/NKX6.3*) replicated in a two-stage analysis (two-stage FDR < 5%) using the DiOGenes study and yielded significant and consistent results using a meta-analysis approach. Bayesian modeling of epigenomic annotation was able to highlight markers in these loci with a possible regulatory impact. Minor alleles from markers nearby *RBGS4* were found associated with increased weight loss. Interestingly, the major allele (C) for rs6981587, near *MIR486/NKX6.3*, was associated with decreased weight loss. Our two cohorts were of European ancestry and the C allele frequency (77%) was consistent with other European populations as well as Asian populations (>70%). However, these allele frequencies were much lower in several African populations (with frequency ranging from 35–46%). This observation would deserve additional follow-up in weight loss studies with participants from different ancestries; as it may have implication for weight loss intervention in these populations, including in admixed populations (e.g. African Americans).

Our analyses highlighted several genes in the vicinity of these association signals. Within the first locus, *RBGS4*, also known as *LINC01363*, encodes a long non-coding RNA (lncRNA) and has not been previously reported to associate with obesity and weight loss interventions. Our current understanding of lncRNA remains limited though this class of RNAs can play an important role in gene regulation[26]. The contribution of lncRNAs as regulators of the endocrine system is widely accepted[27]. Many GWAs have reported association signals in the vicinity of lncRNA loci[28] and gene expression studies reported that those GWA signals had a cis-eQTL effect specific to lncRNA and not to other neighboring

**Table 3 Prioritized SNPs from Bayesian risk variant inference**

| SNP | Chr | Position | Effect allele | Effect allele frequency | Optifast900 | Meta-analysis |
|---|---|---|---|---|---|---|
| rs873822 | 1 | 167126910 | C | 66% | $0.13 \pm 0.04$ ($p = 0.00053$) | $0.14 \pm 0.03$ ($p = 3.43\text{e-}6$) |
| rs870879 | 1 | 167126987 | G | 66% | $0.13 \pm 0.04$ ($p = 0.00068$) | $0.13 \pm 0.03$ ($p = 4.27\text{e-}6$) |
| rs1027493 | 1 | 167132882 | C | 67% | $0.13 \pm 0.04$ ($p = 0.00056$) | $0.13 \pm 0.03$ ($p = 5.06\text{e-}6$) |
| rs6981587 | 8 | 41516915 | C | 77% | $-0.19 \pm 0.05$ ($p = 0.000043$) | $-0.17 \pm 0.03$ ($p = 1.54\text{e-}6$) |

This table present single-SNP association $p$-values for the two cohorts and their meta-analysis for the top risk variant SNPs. Beta coefficients, with their standard error and $p$-value are provided, as estimated by the linear mixed effect model. Positive betas indicate that the effect allele associates with greater weight loss. Source data are provided as a Source Data file

protein-coding genes[29]. In addition, the contribution of lncRNA genes to local gene regulation involves complex processes that are not necessarily linked to the lncRNA transcripts themselves but instead includes processes associated with their production (such as enhancer activities, transcription processes and splicing)[30]. It is also worth noting that lncRNA have been reported as regulators of adipogenesis[31]. Thus, the contribution of lncRNA, such as *RBGS4*, in response to clinical intervention for metabolic diseases may be under-appreciated.

The second locus associating with weight loss encompasses several genes: *MIR486, ANK1, AGPAT6* and *NKX6.3*.

The *MIR486* locus includes two mir genes encoding pre-mir isoforms of the same microRNA (miRNA). MiRNAs emerged as regulators of important biological processes and have been involved in many complex diseases including obesity, insulin resistance, T2D and cancer[32–38]. MiR486 has been shown to regulate *SIRT1* and its expression in human adipose tissue-derived mesenchymal stem cells is controlled by high glucose[39]. Additional studies have shown that miR486 could impact NFKB signaling by inhibition of *NKFB* inhibitors[40] and that miR486 could inhibit *FOXO1*, a key mediator of insulin signaling[41] and triglyceride metabolism[42].

*ANK*1 belongs to the ankyrin family that links membrane proteins to the cytoskeleton and play a large role in cell motility, proliferation and activation. Genetic variants located within *ANK1* have been found associated with glycemic traits, impaired insulin release[65] and T2D onset both in European and Asian populations[43–45]. *ANK1* variants were also shown to have cis-eQTL effect on the expression of small *ANK1* in skeletal muscle[46,47]. In our gene-based analyses, the *ANK1* gene did not emerge as a top candidate owing to the fact that the analysis was based on the full-length gene (283 kb) whilst most of the single-SNP signals were restricted to a 10.2 kb region. Hence the resulting gene-based p-value was influenced by the incorporation of non-associated SNPs masking the effect of associated SNPs.

The third gene located in the vicinity of the identified variants is *AGPAT6* (also known as *GPAT4*, Glycerol-3-Phosphate Acyl-transferase 4). The encoded enzyme catalyzes the conversion of lysophosphatidic acid to phosphatidic acid and thus is an important contributor in TAG biosynthesis. *Agpat6*-deficient mice were shown to have alteration in lipid metabolism in tissue such as adipose and liver[48]. *Agpat6*-deficient mice were also shown to exhibit a 25% reduction in body weight and resistance to both diet-induced and genetically induced obesity[49].

The fourth gene in this locus is *NKX6.3*, member of the NKX family that contribute to numerous developmental processes. In particular, the NKX homeobox 3 gene is involved in the development of the central nervous system, gastro-intestinal tract and pancreas[50]. Interestingly, *NKX6-3* is located close to variants which have been associated with T2D[43]. The recent GWA by Mahajan and colleagues[51] highlighted the rs13262861 SNP (8 kb away from our identified variants) as being associated with T2D. This SNP was not tested in our data as it did not achieve a good imputation metrics ($r^2 < 80\%$). This SNP is known as an *NKX6-3*

cis-eQTL in human islets[52]. Additionally, *NKX6-3* transcripts encode transcription factors required for the development of alpha and beta cells in the pancreas[53] and have been show to influence insulin secretion[54,55]. Glycemic improvements following LCD were only measured in the DiOGenes cohort. Association between our three top *NKX6.3* variants and rs13262861 did not reveal any significant association with insulin sensitivity improvements (Matsuda index). Our prioritization analyses based on epigenomic annotation highlighted rs6981587 as the top regulatory variant. Since these annotation marks are derived from cell lines, deciphering the exact underlying mechanism may be premature and would require access to specific tissues for a subset of our GWA participants (e.g. with liver and fat biopsies). Yet, investigation of the BIOS QTL data[56] found that rs6981587 was an eQTL, in whole blood, of the *NKX6.3, ANK1* and *AGPAT6* genes. Interestingly, only the *NKX6.3* eQTL reached genome-wide significance (FDR 5%), with the rs6981587-T allele associating with decreased expression levels.

To investigate a role for the genes *NKX6.3, ANK1* and *AGPAT6* in metabolic regulation in vivo, we used a knockdown strategy in *Drosophila melanogaster*, as molecular mechanisms controlling fat mass are largely conserved between flies and humans[57]. We used triglyceride content as the main metabolic readout. Triglycerides constitute the major components of lipids[58], and total triglyceride levels are used as a direct measure of fly adiposity[59]. Whole-body knockdown of *NKX6.3/HGTX* led to significant reduction of whole-body triglyceride content. This observation was replicated with independent RNAi hairpins and confirmed using adult-inducible knockdown. *NKX6.3/HGTX* over-expression also led to a reduction of whole-body triglyceride (TAG) content. While this was surprising, this observation is not uncommon in functional screens[60,61] and it suggests that a tight *NKX6.3/HGTX* gene dosage is important to maintain TAG levels.

Tissue-specific inducible knockdown showed that *NKX6.3/HGTX* acts in oenocytes to maintain TAG levels. RNAi knockdown in oenocytes led to decreased TAG levels while overexpression of *NKX6.3/HGTX* led to increased TAG levels. Oenocytes are hepatocyte-like cells and are important to regulate the fly lipid metabolism[59]. Specifically, these cells regulate whole-body TAG level and have a bidirectional metabolic role. Under starvation conditions, oenocytes accumulate lipid droplets; when food is abundant, they regulate growth, development and feeding behavior. This two-way coupling between body fat and oenocytes is analogous to the liver—adipose axis in mammals. While the exact mechanism of how *NKX6.3/HGTX* can control lipid metabolism remains unclear, the effect could be oenocyte-specific, or via inter-organ communications between the fat body, brain and oenocytes.

Our fly results show phenotypic evidence that *NKX6.3* may be involved in lipid metabolism and may thereby contribute to weight loss variation. On the other hand, the published mouse data on *AGPAT6* makes this gene also a plausible candidate. Another possibility is that the associated genetic variant influences all of these genes and that they jointly contribute to the

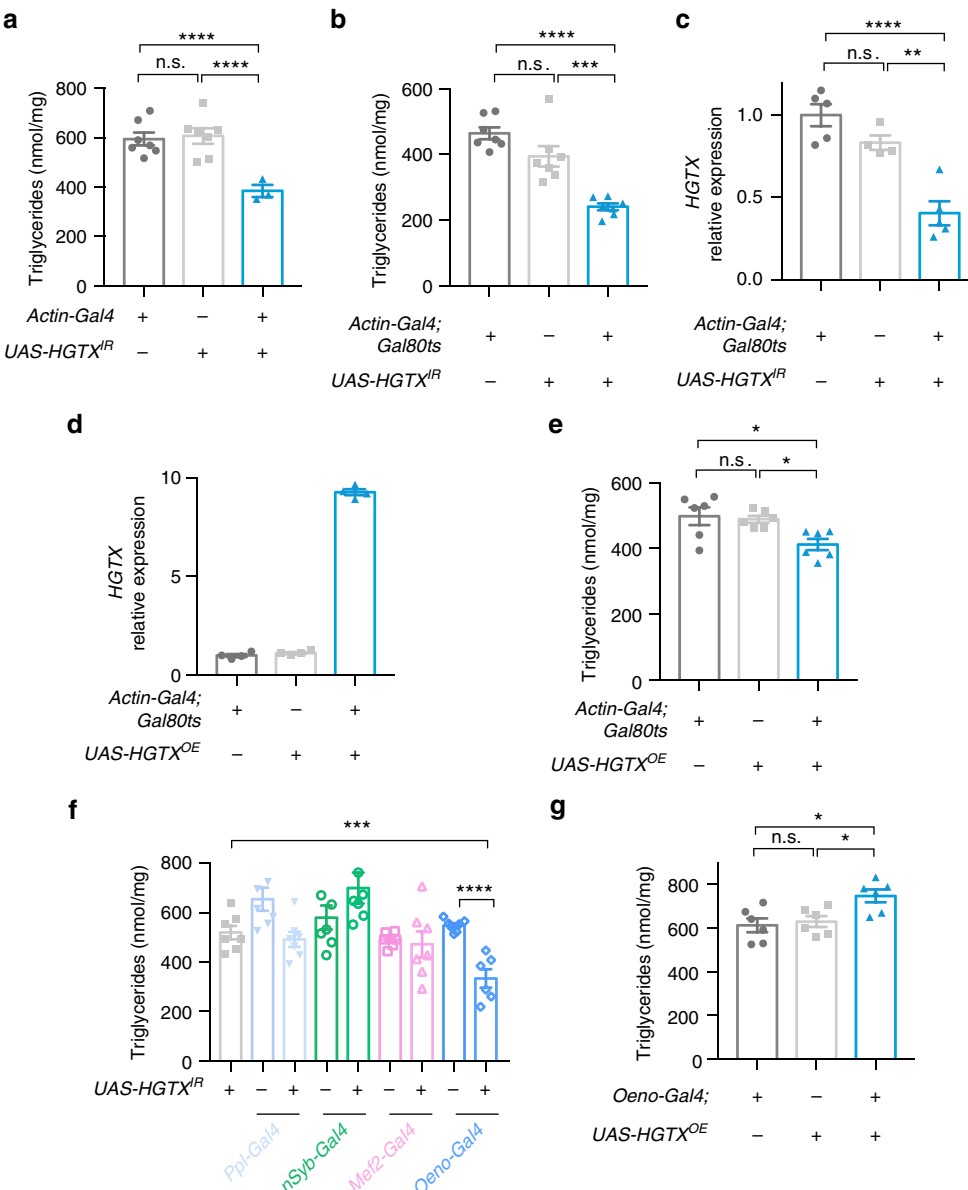

**Fig. 4** *HGTX/NKX6.3* regulates triglyceride (TAG) in *Drosophila*. **a** Whole-body *HGTX* RNAi decreased TAG level in adult flies, $n = 4-8$ groups, 10 flies each. **b** TAG levels in adult inducible of whole-body *HGTX* RNAi flies, $n = 7$ groups, 5 flies each. **c** *HGTX* mRNA was significantly reduced in adult inducible RNAi flies, $n = 4-5$ groups, 5 flies each. **d** *HGTX* mRNA overexpression detected by qPCR, $n = 4$ groups, 5 flies each. **e** TAG levels in adult inducible whole-body *HGTX* over-expression flies, $n = 6$ groups, 5 flies each. **f** TAG levels in tissue-specific *HGTX* RNAi flies, $n = 6-7$ groups, 5 flies each. **g** TAG levels in oenocyte-specific *HGTX* over-expression flies, $n = 6$ groups, 5 flies each. Data are represented as means ± SEM. One-way ANOVA with Bonferroni's multiple comparisons test. $*p < 0.05$, $**p < 0.01$, $***p < 0.001$, $****p < 0.0001$, NS, not significant. Source data are provided as a Source Data file

variability of weight loss in humans. The biological importance of the genes near the weight-loss associated variants and the proximity or overlap with annotated regulatory marks, provides evidence for a functional role in metabolism within this locus. However, given the complex relationship between different organs (brain, adipose, pancreas, liver and muscle) as well as the interplay between metabolic pathways, further investigation would require access to different tissues and under different conditions (e.g. weight loss, induced obesity) to elucidate the potential contribution of each gene.

In conclusion, we performed a weight loss GWA using data from a large clinical practice (the Canadian Optifast900 meal replacement program). Two loci (*RBSG4* and *MIR486/NKX6.3*) were successfully replicated with data from a controlled trial (DiOGenes). Several independent studies provided evidence for a

biological link between the *NKX6-3/MIR486* locus and metabolic disorders including T2D and obesity. This lends additional confidence in our results. Our work opens opportunities for additional functional and preclinical studies to fully elucidate the link between the identified markers and the underlying metabolic and molecular mechanisms.

## Methods

**Ethics**. Local Human Research Ethics Committees, from the Ottawa Hospital and DiOGenes studies, approved the study and all procedures were conducted in accordance with the Declaration of Helsinki. All participants gave informed written consent prior to any testing.

**Study samples**. The Canadian samples consisted of patients enrolled in the Weight Management Clinic[16] who had completed 6 to 12 weeks meal-replacement regimen consisting of a product uniquely available in Canada, Optifast900 (Nestlé

Health Science, Switzerland). Program adherence criteria included: attendance to 75% of clinical sessions and strict adherence to the meal replacement (evaluated by the number of meal-replacement products that were consumed). Patients under medication known to affect rate of weight loss or glucose homeostasis or abnormal thyroid indices were excluded from the analyses. In total, 1436 Optifast900 subjects were eligible for genotyping and had available weight measurements.

The DiOGenes study (NCT00390637) is an interventional, multi-center pan-European study[17,18]. Eight partners participated to the study: Bulgaria, Czech Republic, Denmark, Germany, Greece, the Netherlands, Spain and United Kingdom. Participants followed an 8-week LCD. The LCD provided 800 kcal per day with the use of a meal-replacement product (Modifast, Nutrition et Santé France). Participants could also eat up to 400 g of vegetables (corresponding to a maximal addition of 200 kcal/day). In total, 888 DiOGenes subjects were included for genotyping. The macronutrient composition of the two meal-replacement products was considered similar (fat = 17% and 14%; proteins = 34 and 42% and carbohydrates = 49 and 44%, respectively, for the Optifast and Modifast products).

**Clinical data.** For the Canadian Optifast900 study, weight was measured weekly during the LCD intervention. For the DiOGenes study, weight was measured at baseline and after 1, 3, 5, 7, 8 weeks of LCD. The time point week 5 provided the largest sample size (i.e. smallest percentage of missing values) for both cohorts. Analyses investigated BMI after 5-weeks of LCD, adjusted for baseline BMI, age and gender.

**Genotype data.** Genotype data were generated using HumanCoreExome-12 v1.1 with 264,909 tag SNP marker and 244,593 exome-focused markers. They were processed with the Illumina TM platform following Infinium® HD Assay Ultra, Manual according to manufacturer's instructions. Genotypes were called with the GenomeStudio Software (Illumina). Quality control excluded SNPs with call rate <95%, violating Hardy-Weinberg equilibrium (FDR < 20%), low minor allele frequency <1%. Subjects were excluded if they had low call rate (<95%), abnormally high autosomal heterozygosity (FDR <1%), an XXY karyotype, or gender inconsistencies between genotype data and clinical records. For subjects with high identity-by-state (IBS > 95%), only the one having the highest call rate was kept. Principal component analyses (PCA) were performed independently on each cohort to discard subjects that were outliers in term of genetic structure. Subjects from both cohorts were all of European ancestry and the two cohorts had similar genetic structure (Supplementary Fig. 1). Upon all genetic QCs, 1166 Ottawa and 789 DiOGenes subjects were kept for subsequent analyses.

Genotype imputation was then performed using SHAPEIT[62] and IMPUTE2[63] based on the European reference panel from the 1000 Genome project[64] (March 2012 release, phase 1 version 3). Imputation post-filtering removed SNPs with reference allele frequency less than 1% and INFO score <0.8. Upon such filtering, data for 4.9 M imputed SNPs were available for both datasets.

**Functional analyses in Fly.** Fly strains: Fly stocks were maintained on standard diet with agar, sugar and yeast and were raised in 25 °C incubator at a 12/12 dark and night cycle. $Actin$-$Gal4$ and $TubGal80ts$ was from Bloomington. $UAS$-$Ank^{IR1}$ (GD25945), $UAS$-$HGTX^{IR}$(GD12608), $UAS$-$HGTX^{IR1}$ (KK109732), $UASCG3209^{IR}$ (KK10281) were from the VDRC. $UAS$-$Ank^{OE}$ was from Dr. Ronald R. Dubreuil and $UAS$-$HGTX^{OE}$ (9932) was from Bloomington.

Triglyceride assay: Five male flies were weighted and homogenised in 200 µl PBST (PBS + 0.05% Tween 20) on ice, then sonicated for 10 s using a probe sonicator on ice. After sonication, 800 µl ice-cold PBST was added and mixed thoroughly. Fifty microlitre of the mixture was used to determine the triglycerides using the Roche triglycerides kit (11730711216) under the manufacturer's instructions, and 10 µl of the mixture was used to determine to protein using Bradford protein assay kit (Sigma). Triglycerides were normalized to protein level.

Glycogen assay: Glycogen was determined by measuring the glucose degraded from glycogen using amyoglucosidases. Five male flies were homogenized and dissolved in 1 M KOH solutions. After twice 95% ethanol extraction, the pellet was resuspended 1 ml amyoglucosidase reaction buffer (0.3 mg/ml in 0.25 M acetate buffer, pH 4.75) and incubated in the 37 °C shaking incubator overnight and a glucose assay was performed.

Trehalose assay: Trehalose was determined by measuring the glucose degraded from trehalose using trehalosase. Twenty male flies' heads were homogenized in 200 µl ice-cold TB buffer (5 mM Tris pH 6.6, 136 mM NaCl, 2.7 mM KCL), then centrifuged at 4 °C for 5 min at maximum speed and collect the supernatant. 20 µl of supernatant was used to measure glucose and protein, respectively. Fifty microlitre of supernatant was mixed with 50 µl TE solution (10 µl trehalose in 1.5 ml TB buffer). Incubated at 37 °C for overnight. Centrifuged at 4 °C for max speed for 3 min and then 20 µl of supernatant was used to measure glucose assay.

Glucose assay: 50 µl of mixture was used to determine the [glucose] using Thermo Infinite Glucose kit (TR15421) as the manufacturer's instructions.

Body weight: Body weight was measured by analytical balance.

Food intake assay: Food intake was measured over 24 h using the CAFE assay. Empty vials were employed for evaporation controls. All experiments were set up at Zeitgeber time 6–8, with food intake records starting 24 h after food loading.

Starvation assay: Male flies were put into DAM2 tubes with 2% agar and the death was monitored per hour.

qPCR: 1 µg of mRNA was reverse-transcribed into cDNA using SuperScript® III First-Strand Synthesis System (Invitrogen). All primers used for qPCR that have been prescreened for efficiency and specificity. RT-PCR was performed using Sensimix™ probe kit (Bioline). The program is following: 95 °C 10 min, 40 cycles of 95 °C, 15 s; 55 °C, 15 s; 72 °C, 15 s. The reactions were run on Light Cycle® 480 (Roche). The gene expression was normalized to the reference gene rp49. The following primers are used:

$HGTX$ forward: CGAGTCGCAGGTTAAGGTCT;
$HGTX$ Reverse: CCGCCCATATCGTCCTGTTT;
Rp49 forward: CGGATCGATATGCTAAGCTGT;
Rp49 reverse: GCGCTTGTTCGATCCGTA;
Ilp2 forward: TGAGTATGGTGTGCGAGG;
Ilp2 Reverse: CTCTCCACGATTCCTTGC;
Ilp3 forward: GAACTTTGGACCCCGTGAA;
Ilp3 Reverse: TGAGCATCTGAACCGAACT;
Ilp5 forward: CAAACGAGGCACCTTGGG;
Ilp5 Reverse: AGCTATCCAAATCCGCCA;

Dot blot: Immunoblotting was performed using dot-blot and following a protocol similar to Kim et al.[65]. Ten flies were homogenised in PBS with proteinase inhibitors and debris were centrifuged at 12000 g speed for 2 min, then samples were diluted in 1:200 and 0.4 µl of samples was dropped on the nitrocellulose membrane, a sample without Ilp3 was used as a negative control. Subsequent protocol steps included: block with 5% BSA in TBS-T (0.5–1 h, RT), incubate with primary antibody (0.1–10 µg/ml for purified rabbit-anti-Ilp3 antibody, kindly provided by Dr. Jan Veenstra, Bordeaux University France[66]), 1:500 dissolved in BSA/TBS-T for 30 min at RT. Wash three times with TBS-T (3 × 5 min), incubate with secondary antibody conjugated with HRP for 30 min at RT, wash three times with TBS-T (15 min × 1, 5 min × 2), then once with TBS (5 min), incubate with ECL reagent for 1 min, cover with Saran-wrap. For Coomassie staining, the same amount of supernatant was put into the nitrocellulose membrane and the Coomassie blue staining was dropped into the protein-loaded areas for a few times until no more colour changed, and the membrane was washed once with TBST for imaging. Images were taken using ChemiDoc system (Bio-rad). Quantification was performed using Image J software.

**Statistical analyses.** Single-SNP analyses were performed using BOLT-LMM[24], a Bayesian linear mixed effect model to adjust for population structure and cryptic relatedness between individuals. The rate of weight loss was adjusted for sex, age and starting BMI. Results from the two cohorts, were subsequently meta-analyzed using Genome-Wide Association Meta Analysis (GWAMA) software[67] with random-effect modeling and a double genomic-control (GC) correction[68] (GC correction at study-level and also at meta-analysis level).

Gene-based analysis was performed using the VEGAS (versatile gene-based association study) approach[20]. This method integrates single-SNP p-values at the gene-level and accounts for LD patterns through Monte-Carlo simulations. Analysis was made with the following settings: European ancestry population from the 1000 Genome project as reference population for LD patterns, set-based test using the top 80% SNPs and a gene block size set to 20 kb. Such setting assigns to each gene, its neighboring SNPs (within 20 kb) and discard the 20% SNPs having the least significant p-value. For regions with closely located genes (e.g. gene clusters), potentially the same SNPs could be assign to different genes, leading to different "gene-locus" having similar p-values. This is not an issue as the goal of such analyses is only to highlight loci associated with the trait of interest. Adjustment for multiple-testing was performed using the Benjamini-Hochberg correction[69]. Genome-wide significance threshold was set to FDR < 5%.

Prioritization of GWA signals was performed using a Bayesian framework to model the joint likelihood of association p-values with large-scale epigenomic annotations. Such risk variance inference was performed using the RiVIERA-beta framework[21] and with 450 epigenomic annotations (including histone marks, DNase I hypersensitivity, transcription factor binding, and localization within exons). The goal of this framework is to infer for each input SNP the posterior probability of disease given its association p-value and overlap in functional annotations. Epigenomic annotations were retrieved from Pickrell et al.[70].

Analysis of fly phenotypes was performed using one-way ANOVA with Bonferroni adjustment for multiple comparisons.

**Software.** General statistical analysis was performed using R statistical environment version 3.3.1. We used LocusZoom was used to plot regional association and LD with lead SNP using the 1000 genome CEU population data (hg19/1000 Genomes Mar 2012 EUR). Epigenomic plots were made using the Bioconductor Gviz package. Image J was used to quantified dot-blot results.

**URLs.** For Bolt-LMM, see https://data.broadinstitute.org/alkesgroup/BOLT-LMM/; for GWAMA, see http://www.geenivaramu.ee/en/tools/gwama; for Impute2, see http://mathgen.stats.ox.ac.uk/impute/impute_v2.html; for GenABEL, see http://www.genabel.org/; for Gviz, see https://bioconductor.org/packages/release/bioc/html/Gviz.html; for Riviera-beta, see https://yueli-compbio.github.io/RiVIERA-beta/; for LocusZoom, see http://locuszoom.org/;for Image J, see https://imagej.net.

**Reporting Summary**. Further information on experimental design is available in the Nature Research Reporting Summary linked to this Article.

## Data availability

All summary statistics are freely available (Supplementary Data 1–6). All data that support the findings of this study are available from the corresponding author (Armand.Valsesia@rd.nestle.com) upon reasonable request.

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

## Acknowledgements

We are grateful to Virginie Alexandre for useful scientific discussions and help with project management. We are also grateful to Olivier Simon, Radovan Chytracek, Richard Cote, and Kitsa Paluku for development of a clinical and omics data integration system. We thank Dr. Jan Veenstra for generous gifts of antibodies. The DiOGenes project was supported by the European Commission (Food Quality and Safety Priority of the Sixth Framework Program: FP6-2005- 513946). Local sponsors made financial contributions to the shop centers, which also received a number of foods free of charge from food manufacturers. A full list of these sponsors can be seen at www.DiOGenes-eu.org/sponsors/. Functional analyses in fly were supported by the Australian government NHMRC (National Health and Medical Research Council) project grants APP1028887 and APP1124723. The funders of the study had no role in study design, data collection, data analysis, data interpretation, or writing of the manuscript.

## Author contributions

A.V. and J.H. designed, supervised the study and interpreted results. A.V. designed and led statistical analyses. A.V. performed analyses with contributions from J.C., R.H. and G.L. Q.P.W., V.S., L.J.S. and T.C. performed analyses in flies, under supervision from N.G. and G.N. S.M. and C.C. performed genotyping under supervision from P.D. O.W., P.M. and P.L. managed biological samples. N.V. and D.L. contributed data and biological expertise. W.S. and A.A. designed the DiOGenes clinical study; R.D. and M.E.H. designed the Canadian program. A.V. wrote the manuscript with input from all authors. A.V. had primary responsibility for final content.

## Additional information

**Competing interests:** A.V., J.C., N.G., H.R., G.L., S.M., C.C., O.W., P.M., P.D. and J.H. are full-time employees at Nestlé Institute of Health Sciences SA. D.L. is a member of Institut Universitaire de France. W.H.S. reports having received research support from several food companies such as Nestle, DSM, Unilever, Nutrition et Sante and Danone as well as Pharmaceutical companies such as GSK, Novartis and Novo Nordisk. He is an unpaid scientific advisor for the International Life Science Institute, ILSI Europe. A.A. reports grants or personal fees from McCain Foods, USA, personal fees from McDonald's, USA, personal fees from Basic Research, USA, personal fees from Nestlé, Lausanne, personal fees from Dutch Beer Knowledge Institute, NL, personal fees from Gelesis, USA, personal fees from Novo Nordisk, DK, personal fees from S-Biotek, DK, grants from Arla Foods, DK, grants from Danish Dairy Research Council, grants from Nordea Foundation, DK, outside the submitted work; and Royalties received for the book first published in Danish as Verdens Bedste Kur (Politiken, Copenhagen) and subsequently published in Dutch as Het beste dieet ter wereld (Kosmos Uitgevers, Utrecht/Antwerpen), and in English as World's Best Diet (Penguin, Australia). G.N. is supported by an NHMRC career development fellowship II CDF1111940. The remaining authors declare no competing interests.

