## [Peer Review File · Nature Communications]

Reviewers' comments:

Reviewer #1 (Remarks to the Author):

The manuscript reports findings from GWAS of weight loss in two calorie-restricted diet cohorts. The authors primarily report findings from a gene-based association test in an attempt to increase power in their relatively small samples, with two of the six initial signals replicating in the second cohort. They subsequently perform two orthogonal post-hoc experiments: 1) knockdown of 1 candidate gene in flies, 2) a Bayesian epigenomic marker analysis to prioritize likely candidate functional variants in humans. The fly model proposes a potential candidate gene, ANK1, for weight loss by showing changes in body mass and triglyceride levels after knockdown.

Though a great and growing number of variants have been identified for overall obesity and fat distribution, very few genes have been shown to influence weight loss or weight maintenance. These discoveries could help understand the physiology of weight loss and improve methods for long term maintenance of weight loss, and thus are of great interest to the field.

I see no major flaws in the general methodology of the work. The gene-based test employed as their primary analysis is an established technique that combines evidence from multiple variants to increase power, and their associations appear to be appropriately corrected for multiple testing. Additional presentation of single variant statistics, even just for top variants, or variants in top gene-based signals would be helpful in better assessing the work.

The study presents an interesting and important approach to study obesity and weight change. As these are relatively small studies, and only 1 or 2 genes are suggestively discovered, on their own, these discoveries are unlikely to move the field dramatically. These are challenging studies to do on a large scale, and represent an important approach to addressing the obesity epidemic. Their secondary analyses are less convincing, but are suggestive of both functional candidates and that their discoveries are valid.

Other comments:

1) Though the gene-based method is published and has been used previously, it is not widely known nor well understood. I would recommend moving some of its description into the main text to better aid reader's understanding of the analysis.

2) The two regional plots help illustrate points that are important to the gene-based method that I did not see clearly explained in the methods.

a) How were intergenic variants assigned to genes?

b) As in the densely genotyped region on chr8, how were variants assigned to overlapping genes? Were variants allowed to be included in more than one gene test? If so, particularly for this region what was the count and fraction of overlap in the signals observed for miR486-1/2, NKX6-3, and ANK1?

3) Table 3 highlights candidate functional variants from Riviera-beta. These results and Figure 3, raise several additional questions that are not addressed in the text:

a) Table 3 is the only mention of effect size and direction. First, clarify that positive betas indicate increased weight loss, correct? Second, it seems rather important that the effects for the two discovered loci appear to be in opposite directions, indicating that one suggests resistance to weight loss. This is not mentioned at all in the manuscript, but merits further discussion.

b) The authors present a good summary of the potential role for the discovered loci in obesity and obesity-related diseases, adding strength to their genetic evidence. Some discussion of their potential role in weight loss (and/or calorie restriction), in particular though, seems to be missing, but would be helpful even if more speculative.

c) Riviera-beta is meant to highlight epigenomic evidence for functional alleles, however, figure 3 provides very little information on the evidence derived from this method other than distance to

TSS. Some mention or display of this evidence seems warranted if the authors are going to propose candidate functional variants without additional functional follow up so that readers can better evaluate this evidence. One suggestion may be to plot epigenomic tracks in LocusZoom to show the relationship of these mark to the variants in the region(s).

4) The fly work is a nice addition, but raises three issues.

a) The association data presented don't actually show the evidence for ANK1. The authors mention it wasn't initially a priority gene, but don't really make clear why they chose to test it. This needs to be better demonstrated or explained.

b) As mentioned above for the biological descriptions, the ANK1 evidence only suggests steady-state metabolic changes, not weight loss. This isn't entirely bad, but given the relative lack of overlap discussed in the introduction, providing functional evidence for weight change would be more compelling. Is this possible in flies? Can you do anything to restrict calorie intake and/or measure weight change?

c) The inability to test the other genes in the fly doesn't fully convince that ANK1 is the "functional gene" for the observed association. Would another model have been more appropriate?

Reviewer #2 (Remarks to the Author):

In this GWAS study the authors analyze two cohorts to define the interactions between diet and genetic variation in weight loss. After using a second study to replicate the findings the authors investigate the role of the ANK1 gene, part of the MIR486/NKX6.3/ANK1 locus, in *Drosophila*.

Both the question and the approach are interesting, but the fly work is clearly missing in basic expertise. For example, the authors fail to use proper controls for the *Drosophila* genetics, using only one of the genetic background controls (GAL4/+) instead of also adding the UAS-RNAi/+ control. Moreover, these data are too preliminary and could easily be an artifact of genetic background, therefore the authors should use at least another UAS-RNAi allele. It would also make a stronger case if they added a second ubiquitous driver for their experiments. Finally, given that the authors are using an Actin-GAL4 driver, they should express ANK1 KD only in the adult flies, in order to bypass potential developmental effects.

The results are interesting but very preliminary, for example, the authors find a decrease in TAGs, but an increase in body weight. By looking at the methods, I noticed that the authors homogenized groups of flies in water for their TAG assays; typically, single flies are homogenized in buffer containing detergent for better extraction and quantification. Also, the TAGs are typically normalized to total protein levels to control for body composition. It doesn't look like the authors did not use standard methods used in flies to measure TAGs, so I am unable to draw any conclusions. For example, do protein levels change? Are the flies bloated and full of water? Also, TAGs levels are hardly a proof of metabolic state in flies, and better metabolic measurements are needed to show that indeed these flies are metabolically deregulated (insulin levels, insulin mRNA levels, glycogen levels, etc).

In particular, to show that ANK1 has a role in regulating body fat/composition, the authors should also test the effect of its overexpression- this is standard practice in *Drosophila* studies and I would expect any *in vivo* study to add this. This would allow the authors to ask necessity/sufficiency questions and to make hypotheses about the function of this gene. I would also recommend that the authors target ANK1 knock-down/overexpression to a particular tissue (fat/brain/oenocyte) to see this can recapitulate its effect in a target organ.

In terms of the qPCR studies, the authors should add the sequences of the primers used, their percent efficiency in each sample and whether this was appropriate to apply the differential expression calculations, the Ct number at which the mRNA began to amplify, and the number of technical vs. biological replicates.

Minor issues:

Please fix the syntax errors and misspellings in the manuscript
Figures could be formatted differently to improve clarity

Reviewers' comments:

Reviewer #1 (Remarks to the Author):

The manuscript reports findings from GWAS of weight loss in two calorie-restricted diet cohorts. The authors primarily report findings from a gene-based association test in an attempt to increase power in their relatively small samples, with two of the six initial signals replicating in the second cohort. They subsequently perform two orthogonal post-hoc experiments: 1) knockdown of 1 candidate gene in flies, 2) a Bayesian epigenomic marker analysis to prioritize likely candidate functional variants in humans. The fly model proposes a potential candidate gene, ANK1, for weight loss by showing changes in body mass and triglyceride levels after knockdown.

Though a great and growing number of variants have been identified for overall obesity and fat distribution, very few genes have been shown to influence weight loss or weight maintenance. These discoveries could help understand the physiology of weight loss and improve methods for long term maintenance of weight loss, and thus are of great interest to the field.

I see no major flaws in the general methodology of the work. The gene-based test employed as their primary analysis is an established technique that combines evidence from multiple variants to increase power, and their associations appear to be appropriately corrected for multiple testing. Additional presentation of single variant statistics, even just for top variants, or variants in top gene-based signals would be helpful in better assessing the work.

The study presents an interesting and important approach to study obesity and weight change. As these are relatively small studies, and only 1 or 2 genes are suggestively discovered, on their own, these discoveries are unlikely to move the field dramatically. These are challenging studies to do on a large scale and represent an important approach to addressing the obesity epidemic. Their secondary analyses are less convincing but are suggestive of both functional candidates and that their discoveries are valid.

Response: Thank you for your interest and constructive review. We agree with your remarks on the limitations of the functional studies. In order to address these issues, we have enlarged the scope of the fly screen and the functional annotation. In particular, we have now included tissue specific knock-down data from the fly screen that allow us to also study specific effects for the genes in the region. Indeed, our new results confirm an effect for this region but suggest *NKX6.3* as the most likely candidate gene rather than *ANK1*.

Other comments:

1) Though the gene-based method is published and has been used previously, it is not widely known nor well understood. I would recommend moving some of its description into the main text to better aid reader's understanding of the analysis.

Response: This is a good point, as suggested by the reviewer we provide a short summary in the result section (L61-64):

“To better extract association signals, we used a more elaborate gene-based approach that enables to integrate individual SNP association signals into a single locus-wise signal (see Methods).”

We expand further on the advantages of the gene-based method in the discussion (L163-L177):

“We thus used a well-established gene-based strategy²⁰ and took advantage of recent improvements for single-SNP analyses by using a Bayesian linear-mixed effect model²⁴. Gene-based approaches enable to combine association p-values from individual SNPs into a single locus (e.g. gene-level) p-value. This leads to several benefits. First, statistical evidence is strengthened by integrating association p-values from markers located within a same region. Some of those markers may already display association signals close to genome-wide significance thresholds, yielding a more extreme combined p-value. These approaches account for LD relationships between markers by using resampling approaches (e.g. Monte Carlo simulations). The second benefit from gene-based approaches pertains to a significant reduction of the multiple testing burden (as the number of tests is about 20'000 regions instead of millions of SNPs). Finally, it has been previously discussed that gene-based approaches are less prone to spurious associations caused by population stratification compared to single-SNP or haplotype-based analyses^{19,25}. ”

We also provide more details in the Method section about how intergenic variants are assigned to genes (see response below).

2) The two regional plots help illustrate points that are important to the gene-based method that I did not see clearly explained in the methods.

a) How were intergenic variants assigned to genes?

Response: We clarified this point in the method section. Any variants located within 20kb of a gene would be assigned to it. For gene-clusters, the same variant(s) can be assigned to different genes, this can lead to loci (genes) having similar combined p-value (e.g. *MIR486-1* and *MIR486-2*). Assignments of the same markers to different genes is not an issue, as the goal of gene-based analyses is to highlight regions displaying associations between the encompassed variants and the trait. By no means, it provides a functional evidence that the gene itself is relevant for the trait.

We clarified this point in the Method section (L431-L442):

“ Gene-based analysis was performed using the VEGAS (versatile gene-based association study) approach²⁰. This method integrates single-SNP p-values at the gene-level and accounts for LD patterns through Monte-Carlo simulations. Analysis was made with the

following settings: European ancestry population from the 1000 Genome project as reference population for LD patterns, set-based test using the top 80% SNPs and a gene block size set to 20kb. Such setting assigns to each gene, its neighboring SNPs (within 20kb) and discard the 20% SNPs having the least significant p-value. For regions with closely located genes (*e.g.* gene clusters), potentially the same SNPs could be assigned to different genes, leading to different “gene-locus” having similar p-values. This is not an issue as the goal of such analyses is only to highlight loci associated with the trait of interest.”

b) As in the densely genotyped region on chr8, how were variants assigned to overlapping genes? Were variants allowed to be included in more than one gene test? If so, particularly for this region what was the count and fraction of overlap in the signals observed for miR486-1/2, NKX6-3, and ANK1?

Response: Indeed, gene-based results for *MIR486-1/2* and *NKX6.3* are based on essentially the same SNPs (due to the assignments of variants within 20kb of the gene coordinates, see above response). By contrast, *ANK1* includes additional, and non-significant SNPs, which makes its gene-based pvalue bigger (“less significant”).

This point is addressed in the discussion (L226-231): “In our gene-based analyses, the *ANK1* gene did not emerge as a top candidate owing to the fact that the analysis was based on the full-length gene (283kb) whilst most of the single-SNP signals were restricted to a 10.2kb region. Hence the resulting gene-based p-value was influenced by the incorporation of non-associated SNPs masking the effect of associated SNPs. “

3) Table 3 highlights candidate functional variants from Riviera-beta. These results and Figure 3, raise several additional questions that are not addressed in the text:

a) Table 3 is the only mention of effect size and direction. First, clarify that positive betas indicate increased weight loss, correct? Second, it seems rather important that the effects for the two discovered loci appear to be in opposite directions, indicating that one suggests resistance to weight loss. This is not mentioned at all in the manuscript, but merits further discussion.

Response: Thank you for raising this interesting point. We clarify in the caption that positive betas correspond to increased weight loss. The *MIR486/NKX6.3* SNP has a negative beta, indicating that the C allele (major allele) associates with lesser weight loss. Interestingly, the C allele frequencies (77%) from our cohorts are consistent with other European populations as well as with Asian populations (with frequency > 70%). However, differences were found with several African populations, where the C allele was found as the minor allele (and with frequencies ranging from 35% to 46%).

This indeed merits further discussion, as it may have implication regarding the allele prevalence and possible weight loss response (proportion of responders; and average population weight loss) in African and admixed populations (such as African-American that are severely affected by obesity). However, in the absence of weight loss data in subjects

from African ancestries and given the large inter-subject variability to weight loss, we cannot comment further.

We included this point in our discussion (L183-193):

“Minor alleles from markers nearby *RBGS4* were found associated with increased weight loss. Interestingly, the major allele (C) for rs6981587, near *MIR486/NKX6.3*, was associated with decreased weight loss. Our two cohorts were of European ancestry and the C allele frequency (77%) was consistent with other European populations as well as Asian populations (>70%). However, these allele frequencies were much lower in several African populations (with frequency ranging from 35% to 46%). This observation would deserve additional follow-up in weight loss studies with participants from different ancestries; as it may have implication for weight loss intervention in these populations, including in admixed populations (e.g. African Americans).”

b) The authors present a good summary of the potential role for the discovered loci in obesity and obesity-related diseases, adding strength to their genetic evidence. Some discussion of their potential role in weight loss (and/or calorie restriction), in particular though, seems to be missing, but would be helpful even if more speculative.

Response: Our new fly data highlight a potential impact of *NKX6.3* in insulin signalling and TAG biosynthesis with an effect in fly oenocytes (hepatocyte-like cells). The observed phenotype seems consistent with the known physiology and role of oenocytes. But the exact *NKX6.3* mechanism in humans and the physiological role in weight loss is not known. Thus, as mentioned by the reviewer the role for this gene in weight loss and in particular how it influences the interplay between different organs (liver, adipose, brain, and pancreas) are rather speculative at this point. However, we have added these aspects in our discussion (L264-297).

“Tissue-specific inducible knockdown showed that *NKX6.3/HGTX* acts in oenocytes to maintain TAG levels. Oenocytes are hepatocyte-like cells and are important to regulate the fly lipid metabolism⁶⁰. Specifically, these cells regulate whole-body TAG level and have a bidirectional metabolic role. Under starvation conditions, oenocytes accumulate lipid droplets; when food is abundant, they regulate growth, development and feeding behavior. This two-way coupling between body fat and oenocytes is analogous to the liver – adipose axis in mammals. While the exact mechanism of how *NKX6.3/HGTX* can control lipid metabolism remains unclear, the effect could be oenocyte-specific, or via inter-organ communications between the fat body, brain and oenocytes. Lending support to the latter hypothesis, *NKX6.3/HGTX* knockdown led to alteration of Insulin-like peptide 3 (Ilp3) levels. Ilp3 is secreted from insulin-producing cells (IPCs) in the brain⁶¹. IPCs play a key endocrine secretion role in fly and express genes homologous to those from β -cells of the mammalian pancreas^{61,62}. Ilp3 has been associated with energy homeostasis, TAG alteration and metabolic defects^{63,64}. The *dilp3* (drosophila insulin-like peptide 3) gene expression is nutrient-sensitive, regulated by complex cis-regulatory elements and its expression appears independent from other *dilp* genes⁶². This may explain the fact that only Ilp3 is downregulated and not Ilp2 and Ilp5. It could also mean either a compensation mechanism,

or alternatively could indicate participation of the intestine, as *Ilp3* is also highly expressed in the fly gut.

Our fly results show phenotypic evidence that *NKX6.3* may be involved in lipid metabolism and may thereby contribute to weight loss variation. On the other hand, the published mouse data on *AGPAT6* makes this gene also a plausible candidate. Another possibility is that the associated genetic variant influences all of these genes and that they jointly contribute to the variability of weight loss in humans. The biological importance of the genes near the weight-loss associated variants and the proximity or overlap with annotated regulatory marks, provides evidence for a functional role in metabolism within this locus. However, given the complex relationship between different organs (brain, adipose, pancreas, liver and muscle) as well as the interplay between metabolic pathways (insulin signaling and TAG biosynthesis), further investigation would require access to different tissues and under different conditions (e.g. weight loss, induced obesity) to elucidate the potential contribution of each gene. ”

c) Riviera-beta is meant to highlight epigenomic evidence for functional alleles, however, figure 3 provides very little information on the evidence derived from this method other than distance to TSS. Some mention or display of this evidence seems warranted if the authors are going to propose candidate functional variants without additional functional follow up so that readers can better evaluate this evidence. One suggestion may be to plot epigenomic tracks in LocusZoom to show the relationship of these mark to the variants in the region(s).

Response: We now provide a new figure 3, that better displays the underlying epigenomic annotation, jointly with association p-values and PPA for tested variants. Since our epigenomic modelling is based on more than 450 annotation marks, displaying all tracks was not convenient. Thus, we collapsed the different DNase tracks into a single one and we present individually other marks (e.g. histones). Also constrained by the amount of annotation, we limited these plots to variants that were used in the Riviera-beta analyses.

4) The fly work is a nice addition but raises three issues.

a) The association data presented don't actually show the evidence for ANK1. The authors mention it wasn't initially a priority gene, but don't really make clear why they chose to test it. This needs to be better demonstrated or explained.

Response: The fly screen relies on two main conditions: a) an ortholog of the human gene with sufficiently high similarity and b) the full-body knock-down of that gene being viable and reproductive. All genes that are within the association region for which orthologs exist in the fly were tested. At the time of submission only the *ANK1* gene knock-down showed sufficient survival and an interesting effect on TAGs. Improving our screen to reduce mortality and performing tissue-specific knock-down, we now provide a more comprehensive assessment of nearby genes including new data on *NKX6.3* and *AGPAT6*. The two RNA genes (*MIR486* and *RBSG4*) are not conserved in fly and could not be tested.

All these results are now shown in the section “Functional assessment of ANK1, AGPAT6 and NKX6.3 in Drosophila melanogaster” (L103-139).

b) As mentioned above for the biological descriptions, the ANK1 evidence only suggests steady-state metabolic changes, not weight loss. This isn't entirely bad but given the relative lack of overlap discussed in the introduction, providing functional evidence for weight change would be more compelling. Is this possible in flies? Can you do anything to restrict calorie intake and/or measure weight change?

Response: The body weight change in flies can be detected by analytical balance. We have measured the body weight change after starvation (in wild-types) and could see ~23% decrease in body weight (see adjacent figure).

In our new data, we measured body weight for knock down animals, we did not see a significant difference when compared to both parental controls. Body weight is not traditionally used as a metabolic readout for flies, as both body size and the presence of an exoskeleton can complicate the interpretation of these types of measurements. By contrast, TAGs are commonly used as a direct measure of fly adiposity (Pospisilik et al. Cell 2010), thus we decided to focus on this readout and incorporated additional metabolic readouts (see responses to reviewer #2).

c) The inability to test the other genes in the fly doesn't fully convince that ANK1 is the "functional gene" for the observed association. Would another model have been more appropriate?

Response: The GWAS data provides a location for the association signal but does not allow to identify the gene responsible for the signal. As such all genes within the region are potential positional candidates. *Drosophila* as a first *in vivo* screening tool allows to quickly evaluate potential phenotypic consequences for many genes in a GWAS region, provided that reasonable orthologous genes exist in the fly. This model can help to select the most likely candidate gene in a region. The potential of this approach has been shown with several studies (Neely et al. Cell 2010; Pospisilik et al. Cell 2010; Yamamoto et al. Cell 2014). Follow-up of results from human GWAS is still comparatively new. Nevertheless, the fly organism holds significant potential (Wangler et al. Dis Model Mech. 2017); as recently demonstrated for BMI GWAS signals (Baranski et al. PLoS Genetics 2018).

We therefore believe that the fly is a useful model as it is easy to manipulate genetically, and triglyceride measurements provide a good readout for effect of knockdown on lipid metabolism. Even though mouse models might be more appropriate, it is too time-consuming as a discovery tool to test multiple genes.

Fig.: Body weight reduced after 24h starvation. Mean \pm S.E.M. t test, $p < 0.01$

Reviewer #2 (Remarks to the Author):

In this GWAS study the authors analyze two cohorts to define the interactions between diet and genetic variation in weight loss. After using a second study to replicate the findings the authors investigate the role of the ANK1 gene, part of the MIR486/NKX6.3/ANK1 locus, in *Drosophila*. Both the question and the approach are interesting, but the fly work is clearly missing in basic expertise.

For example, the authors fail to use proper controls for the *Drosophila* genetics, using only one of the genetic background controls (*GAL4/+*) instead of also adding the *UAS-RNAi/+* control. Moreover, these data are too preliminary and could easily be an artifact of genetic background, therefore the authors should use at least another *UAS-RNAi* allele.

Response: The reviewer makes an important point here. The data originally included in this manuscript was part of a larger screening result (from multiple independent projects), and so at the time of submission we had not performed individual *UAS-RNAi/+* controls, as we have not previously seen a strong phenotype with the parental RNAi/+ control.

During revisions for this manuscript, we have tested *Ank/ANK1* with the additional control *UAS-Ank RNAi/+* as suggested. We found that triglycerides in *Act-Gal4>UAS-Ank RNAi* animals are not significantly reduced when compared *UAS-Ank RNAi/+* control and we have repeated this with an additional hairpin; using an adult-inducible knock-down, as well as tissue-specific knock-down (see below figure). We have not seen such a strong effect from the parental RNAi/+ control when studying other VDRC lines.

Taken together our data no longer supports a clear-cut role for *Ank* in regulating TAG levels, and we have acknowledged these results.

We thank the reviewer for pointing out this issue, and we believe this has helped us make a much stronger manuscript.

Although the new data suggest that *ANK1* knock-down may not show a specific phenotype, using the approach as suggested by the reviewer we include new data, to test other nearby genes (*AGPAT6* and *NKX6.3*). The *AGPAT6* fly ortholog knock-down does not show any significant phenotype (with 4 x 10 independent biological replicates).

However, we do observe a very strong difference in TAG levels for *HGTX/NKX6.3* knock down animals when compared to parental controls (**Figure 4A and Figure S1A**), both using a developmental knock down and also adult-specific inducible knock down (**Figure 4B**). As a whole our data supports a role for adult *HGTX* in regulating TAG levels, and additional genetic experiments show this is through actions in the fly oenocytes (**Figure 4E**). We have thus performed all of the requested revision experiments with a focus on *HGTX/ NKX6-3* (see the new results section entitled “**Functional assessment of ANK1, AGPAT6 and NKX6.3 in *Drosophila melanogaster***”, L103-139).

It would also make a stronger case if they added a second ubiquitous driver for their experiments.

Response: We have tested two more ubiquitous drivers (*Tub-Gal4* and *DA-Gal4*) to perform *HGTX/ NKX6-3* RNAi knockdown. However, both drivers caused 100% lethal in F1 progenies, whereas we had 95% lethality with the *Act-Gal4* driver.

Finally, given that the authors are using an Actin-GAL4 driver, they should express ANK1 KD only in the adult flies, in order to bypass potential developmental effects.

Response: We appreciate the reviewers concerns and have now performed the suggested adult inducible knock down experiments for *HGTX/ NKX6-3* using a *TubGal80ts* strategy. Adult inducible knock down animals still display a strong reduction in TAG levels compared to both parental controls (**Figure 4A-B and Figure S2A**).

The results are interesting but very preliminary, for example, the authors find a decrease in TAGs, but an increase in body weight. By looking at the methods, I noticed that the authors homogenized groups of flies in water for their TAG assays; typically, single flies are homogenized in buffer containing detergent for better extraction and quantification. Also, the TAGs are typically normalized to total protein levels to control for body composition. It doesn't look like the authors did not use standard methods used in flies to measure TAGs, so I am unable to draw any conclusions. For example, do protein levels change? Are the flies bloated and full of water?

Response: With the new data on *HGTX* RNAi flies, we did not observe any difference in body weight. In addition, in the revised manuscript, we have used the standard protocol for all TAG assays (Tennessee et al. Methods 2014) to confirm the results. We use PBST (PBS + 0.05% Tween 20) for TAG extraction and measure the protein level at same time. The TAG level is normalized to protein level in each sample.

We updated the method accordingly (L370-376):

“Triglyceride assay: 5 male flies were weighted and homogenised in 200 µl PBST (PBS + 0.05% Tween 20) on ice, then sonicated for 10s using a probe sonicator on ice. After sonication, 800µl ice-cold PBST was added and mixed thoroughly. 50 µl of the mixture was used to determine the triglycerides using the Roche triglycerides kit (11730711216) under the manufacturer’s instructions, and 10 µl of the mixture was used to determine to protein using Bradford protein assay kit (Sigma). Triglycerides were normalized to protein level.”

Also, TAGs levels are hardly a proof of metabolic state in flies, and better metabolic measurements are needed to show that indeed these flies are metabolically deregulated (insulin levels, insulin mRNA levels, glycogen levels, etc).

Response: In the revised manuscript, we have included more metabolic measurements. We did not see significant changes in glycogen, trehalose, body weight, food intake, and starvation response (Figure S2B-F). We also investigated mRNA levels for insulin-like peptide 1 (*Ilp2*), insulin-like peptide 3 (*Ilp3*) and insulin-like peptide 5(*Ilp5*) using qPCR. We found that *Ilp3* mRNA is dramatically reduced (Figure 4C).

In particular, to show that ANK1 has a role in regulating body fat/composition, the authors should also test the effect of its overexpression- this is standard practice in Drosophila studies and I would expect any in vivo study to add this. This would allow the authors to ask necessity/sufficiency questions and to make hypotheses about the function of this gene.

Response: We appreciate the reviewers comments here. We have tried over-expression studies with *ANK1* and did not see a phenotype, consistent with *ANK1* playing no role in TAG regulation (see adjacent figure).

Fig: TAG does not change in Ank/ANK1 overexpression flies. Mean ± S.E.M. One-way ANOVA with Tukey test test, n.s. not

We also performed *HGTX/NKX6.3* overexpression studies with multiple ubiquitous drivers *Actin-Gal4*, *DA-gal4* and *Tub-Gal4*, however in each case this manipulation was 100% lethal.

I would also recommend that the authors target ANK1 knock-down/overexpression to a particular tissue (fat/brain/oenocyte) to see this can recapitulate its effect in a target organ.

Response: We thank the reviewer for this helpful suggestion.

The focus of our revised manuscript is now on *HGTX/NKX6.3*; and we performed these experiments to identify the specific compartment responsible for the observed phenotype.

We have tested *HGTX/NKX6.3* RNAi in the fat body (*Ppl-Gal4*), muscle (*Mef2-Gal4*), brain (*nSyb-Gal4*) and oenocytes (*Oneo-Gal4*) and found the specific loss of *HGTX* in oenocytes resulted in reduce TAG level when compared to parental control lines (**Figure 4E**).

In terms of the qPCR studies, the authors should add the sequences of the primers used, their percent efficiency in each sample and whether this was appropriate to apply the differential expression calculations, the Ct number at which the mRNA began to amplify, and the number of technical vs. biological replicates.

Response: in the previous *Ank/ANK1* qPCR test, we used 4 biological samples for each genotype and two pairs of primers. Both primers gave similar results and one representative result was shown.

In the revised manuscript for *HGTX/NKX6.3* qPCR assay, we used 5 biological samples for each genotype and two pairs of primers. Both primers gave similar results and one representative result was shown (**Figure 4C**). The details of qPCR (including primer sequences) were put in the method section (L403-414).

This now reads as

“qPCR 1µg of mRNA was reversely transcribed into cDNA using SuperScript® III First-Strand Synthesis System (Invitrogen). All primers used for qPCR that have been prescreened for efficiency and specificity. RT-PCR was performed on Light Cycler® 480 (Roche) with the Sensimix™ probe kit (Bioline), and the following cycling conditions: 95°C 10 min, 40 cycles of 95°C, 15s; 55°C, 15s; 72°C, 15s. The reactions were run on Light Cycle® 480 (Roche). The gene expression was normalized to the reference gene rp49. The following primers are used:

- *HGTX* forward: CGAGTCGCAGGTTAAGGTCT;
- *HGTX* Reverse: CCGCCCATATCGTCCTGTTT;
- *Rp49* forward: CGGATCGATATGCTAAGCTGT;
- *Rp49* reverse: GCGCTTGTTTCGATCCGTA

“

Minor issues:

Please fix the syntax errors and misspellings in the manuscript

Response: our manuscript has been checked and corrected accordingly.

Figures could be formatted differently to improve clarity

Response: We provided high resolution figures in the revised manuscript, and a new figure 3 that simplifies the display of the Bayesian epigenomic modelling.

Reviewers' comments:

Reviewer #1 (Remarks to the Author):

This revised manuscript highlighting discovery of two regions associated with genetic control of response to weight loss intervention is much improved. The authors were attentive to the comments and critiques of the reviewers, yielding more useful and interpretable results.

While the underlying human genetic data appear to be unchanged from the original manuscript, the presentation is improved, both in figures and text, and the improved, more rigorous approach to the fly experiments make the manuscript more compelling than the original, and substantially improve the interpretation of this work. It is unfortunate that similar analyses in model systems could not be undertaken for both associated loci.

Addressing the potential function of all the candidates in the region was of paramount importance, as highlighted by the change in the focus (and resulting "effector gene" of the fly results. It appears that the substantial additions and changes to the fly analyses also have made substantial improvements to the functional model assays.

The inclusion of epigenomic marks in the figure helps to make those results more interpretable. Assuming the fly results are correct, can the authors use that information and the epigenomic data to suggest a mechanism for how rs6981587 may be affecting NKX6.3 in humans?

Reviewer #2 (Remarks to the Author):

The authors did a satisfactory job at addressing nearly all of the points I raised in the initial review. It is relieving to see that the addition of proper controls identified false positive phenotypes in ANK1, and that the authors applied the suggested approach to the study of NKX6. Overall, these experiments place NKX6 at the center of an interesting phenotype that needs to be further characterized in the future and that may be outside the scope of the manuscript. I only have two major comments, detailed below.

Originally, I argued that the authors should test ANK1 overexpression. Since we don't know the effects of these SNPs on the transcript levels or protein function, overexpression experiments should be carried out. The authors report that overexpression of NKX6 was lethal. This isn't surprising, but as it was done with RNAi in Figure 4, the authors should try to limit expression of this gene only in adults and with the oenocyte GAL4. I think this experiment would strengthen the study and should be done.

My main concern is on the claims about changes in dilp3. The qPCR data for this transcript are weak- clearly the UAS-HGTX IR has an insulin phenotype on its own because of the genetic background. Antibodies to dILP3 are available and should be used to test the authors' claim that that NKX6 leads to "alterations of Ilp3." The experiments are quite straightforward and a more direct and satisfactory test of the hypothesis that Ilp3 and its secretion are changed by this genetic manipulation of NKX6. Weak changes in transcripts that could be attributable to background effects and of unknown origin (brain vs, gut), significantly weaken this figure.

Minor comments:

Oenocyte-GAL4 is sometimes abbreviated as "oeno" and sometimes "oneo"

Figure 4 legend has two "C" panels

Still a few spelling and grammatical mistakes in the text

Reviewers' comments:

Reviewer #1 (Remarks to the Author):

This revised manuscript highlighting discovery of two regions associated with genetic control of response to weight loss intervention is much improved. The authors were attentive to the comments and critiques of the reviewers, yielding more useful and interpretable results.

While the underlying human genetic data appear to be unchanged from the original manuscript, the presentation is improved, both in figures and text, and the improved, more rigorous approach to the fly experiments make the manuscript more compelling than the original, and substantially improve the interpretation of this work. It is unfortunate that similar analyses in model systems could not be undertaken for both associated loci.

Addressing the potential function of all the candidates in the region was of paramount importance, as highlighted by the change in the focus (and resulting "effector gene" of the fly results. It appears that the substantial additions and changes to the fly analyses also have made substantial improvements to the functional model assays.

Response:

Thank you for your review and positive comments.

The inclusion of epigenomic marks in the figure helps to make those results more interpretable. Assuming the fly results are correct, can the authors use that information and the epigenomic data to suggest a mechanism for how rs6981587 may be affecting NKX6.3 in humans?

Response:

The epigenomic marks are derived from cell line experiments (ENCODE project). In GWA studies, these marks are commonly used for variant prioritization, but it may be premature to propose a mechanism based on such annotation.

Nevertheless, the overlap with DNase sensitivity cluster and several repression marks suggest a gene expression regulatory mechanism.

And indeed, the recent study by the BIOS QTL consortium (Zhernakova et al, Nature Genetics 2017) found that rs6981587 was an eQTL for *NKX6.3* ($p=4.34e-42$), *AGPAT6* ($p=1.2e-8$) and *ANK1* ($p=2.5e-6$); with only *NKX6.3* reaching genome-wide significant level (FDR 5%). Specifically, they found that the rs6981587-T allele was associated with lower *NKX6.3* levels (allele Z-score -3.93).

We included these observations in our discussion (L265-273):

“Our prioritization analyses based on epigenomic annotation highlighted rs6981587 as the top regulatory variant. Since these annotation marks are derived from cell lines, deciphering the exact underlying mechanism may be premature and would require access to specific tissues for a subset of our GWA participants (e.g. with liver and fat biopsies). Yet, investigation of the BIOS QTL data⁵⁶ found that rs6981587 was an eQTL, in whole blood, of the *NKX6.3*, *ANK1* and *AGPAT6* genes. Interestingly, only the *NKX6.3* eQTL reached genome-wide significance (FDR 5%), with the rs6981587-T allele associating with decreased expression levels.”

Reviewer #2 (Remarks to the Author):

The authors did a satisfactory job at addressing nearly all of the points I raised in the initial review. It is relieving to see that the addition of proper controls identified false positive phenotypes in ANK1, and that the authors applied the suggested approach to the study of NKX6. Overall, these experiments place NKX6 at the centre of an interesting phenotype that needs to be further characterized in the future and that may be outside the scope of the manuscript. I only have two major comments, detailed below.

Response: Thank you very much for your new review and your very useful comments. We have performed the requested experiments. We believe these new data strengthen our study and we provide below a point-by-point response.

Q1: Originally, I argued that the authors should test ANK1 overexpression. Since we don't know the effects of these SNPs on the transcript levels or protein function, overexpression experiments should be carried out.

Response: We have now included over-expression experiments of ANK1 using a whole-body driver (*Actin-Gal4*). There is no impact on triglyceride levels in the *Actin-Gal4>UAS-ANK1 OE* animals compared to control (Figure S2).

These negative results have been added L111-113:

“We also performed over-expression (OE) of ANK1 using a whole-body driver (*Actin-Gal4*). There was no impact on triglyceride levels in the *Actin-Gal4>UAS-ANK1 OE* animals compared to controls (Figure S2).”

The authors report that overexpression of NKX6 was lethal. This isn't surprising, but as it was done with RNAi in Figure 4, the authors should try to limit expression of this gene only in adults and with the oenocyte GAL4. I think this experiment would strengthen the study and should be done.

Response: We now have driven overexpression of *HGTX/NKX6.3* in the adult using *UAS-Gal80ts*. The *HGTX* is over-expressed for 6 days in adults and then triglycerides and insulin-like peptides were measured. The *HGTX* mRNA expression is ~9 times higher in *Actin-Gal4; Gal80ts>UAS- HGTX OE flies* (Figure 4D). We saw a slight reduction in TAG levels when we overexpress *HGTX* ubiquitously (Figure 4E).

We also performed oenocyte-specific overexpression of *HGTX* and observed a significant increase in TAG, thereby reversing the phenotype observed with oenocyte-specific knock down.

We present these data at L137-142:

“To further confirm the role of *HGTX* in regulation of TAG, we used inducible over-expression of *HGTX* in adults with mRNA expression ~9 times higher in *Actin-Gal4; Gal80ts>UAS-HGTX OE* animals (Figure 4D) compared to the parental controls. *HGTX* over-expression led to a mild reduction in TAG (Figure 4E). No significant impact was observed for *Ilp2*, 3 and 5 mRNA expression or *dilp3* protein levels (Supplementary Fig. 5).

And also, L147-150:

“Conversely, oenocyte-specific overexpression of HGTX resulted in a significant increase in TAG (Figure 4G). Together, our data supports a role for HGTX/NKX6.3 acting in the fly oenocyte to regulate triglyceride content in vivo.”

And finally, we discuss these new OE results (L279-285):

“Whole-body knockdown of NKX6.3/HGTX led to significant reduction of triglyceride content. This observation was replicated with independent RNAi hairpins and confirmed using adult-inducible knockdown. NKX6.3/HGTX over-expression also led to a reduction of whole-body triglyceride (TAG) content. While this was surprising, this observation is not uncommon in functional screens^{60,61} and it suggests that a tight NKX6.3/HGTX gene dosage is important to maintain TAG levels. “

We believe that these new data strengthen the possible role of NKX6.3/HGTX in modulating lipid metabolism. However, we acknowledge that the exact underlying mechanisms would deserve a dedicated follow-up study, that is beyond the scope of the current study.

My main concern is on the claims about changes in dilp3. The qPCR data for this transcript are weak- clearly the UAS-HGTX IR has an insulin phenotype on its own because of the genetic background. Antibodies to dILP3 are available and should be used to test the authors' claim that that NKX6 leads to “alterations of Ilp3.” The experiments are quite straightforward and a more direct and satisfactory test of the hypothesis that Ilp3 and its secretion are changed by this genetic manipulation of NKX6. Weak changes in transcripts that could be attributable to background effects and of unknown origin (brain vs, gut), significantly weaken this figure.

Response: This is a very good point and we are thankful to the reviewer for mentioning it. By qPCR analysis, we observed *Ilp3* mRNA expression was significantly reduced in inducible RNAi flies. Following the reviewer’s comment, we assessed Ilp3 protein levels by dot-blot (see Figure S4 and S5 for RNAi and OE assays respectively). However, we did not observe any significant change in Ilp3 protein levels.

We present these data at L133-136 for the RNAi assays:

“Of note, HGTX inducible knockdown did not affect fly insulin-like peptide Ilp2 or Ilp5 expression but resulted in a decrease in Ilp3 expression. However, we did not observe any difference in dilp3 expression at the protein level (Supplementary Fig. 4).

We also tested mRNA and protein levels in the OE assays, but we did not observe any significant changes. These data are included L140-142:

“No significant impact was observed for Ilp2, 3 and 5 mRNA expression or dilp3 protein levels (Supplementary Fig. 5).

In the whole manuscript, we removed accordingly all mentions of impact on the insulin phenotypes.

Minor comments:

Oenocyte-GAL4 is sometimes abbreviated as “oeno” and sometimes “oneo”

Response: We have gone through all text and corrected this mistake

Figure 4 legend has two “C” panels

Response: We have rearranged the data and corrected this mistake.

Still a few spelling and grammatical mistakes in the text

Response: We have substantially revised the text and removed spelling or grammatical errors.

REVIEWERS' COMMENTS:

Reviewer #2 (Remarks to the Author):

I am satisfied by the experiments and changes the authors carried out at the suggestions of the reviewers and I believe this is a much stronger and balanced manuscript after the peer review process.

REVIEWERS' COMMENTS:

Reviewer #2 (Remarks to the Author):

I am satisfied by the experiments and changes the authors carried out at the suggestions of the reviewers and I believe this is a much stronger and balanced manuscript after the peer review process.

Response: Thank you very much for your thorough review and useful comments.